# Impacts of Dietary Macronutrient Pattern on Adolescent Body Composition and Metabolic Risk: Current and Future Health Status—A Narrative Review

**DOI:** 10.3390/nu12123722

**Published:** 2020-12-02

**Authors:** Oh Yoen Kim, Eun Mi Kim, Sochung Chung

**Affiliations:** 1Department of Health Science, Graduate School, Dong-A University, Busan 49315, Korea; oykim@dau.ac.kr; 2Department of Food Science and Nutrition, Dong-A University, Busan 49315, Korea; 3Dietetic Department, Kangbuk Samsung Hospital, Seoul 03181, Korea; em82.kim@samsung.com; 4Department of Pediatrics, Konkuk University Medical Center, Konkuk University School of Medicine, Seoul 05030, Korea

**Keywords:** adolescent obesity, dietary macronutrient modification, body composition, metabolic status

## Abstract

Obesity, particularly in childhood and adolescence, is one of the serious public health problems worldwide. According to the World Health Organization, 10% of young people aged 5–17 years are obese, which is rapidly increasing around the world. Furthermore, approximately 80% of adolescents who become obese develop bodyweight-related health problems in adulthood. Eating habits and lifestyles play important roles in forming body composition and metabolic status. Changes in body composition in adolescence, the period in which secondary sex characteristics begin to develop, can alter hormonal and metabolic status, can consequently affect health status and the risk of developing chronic diseases in adulthood, and moreover may have an impact on probable body composition and metabolic status in the next generation. Here, we reviewed cross-sectional and interventional studies to analyze the role of dietary patterns focusing on macronutrient intake in growth, body composition, and metabolic changes in adolescents. These findings provide insights into optimal dietary guidelines for healthy growth with accretion of adequate body composition in adolescence, and provide an effective strategy for preventing and managing the risk of obesity-related metabolic disease in adulthood, with the additional benefit of providing potential benefits for the next generation’s health.

## 1. Introduction

The prevalence ratios of overweight and obesity continue to increase worldwide, with subsequent increases in the burden of health care costs [1,2]. Obesity in childhood and adolescence in particular is thought to be one of most serious global public health problems in the 21st century [3]. According to a World Health Organization (WHO) European report from 2017, 10% of children and adolescents aged 5 to 17 years are obese, with rapidly increasing proportions in many countries and regions [4]. Up to 80% of obese adolescents may develop bodyweight-related health problems that are traditionally more common in adulthood, such as metabolic syndrome, type 2 diabetes, sleep apnea, and cardiovascular disease [4,5,6].

Various behavioral and emotional aspects are related to the occurrence of obesity, which affects adolescents’ quality of life [7,8,9]. In particular, dietary factors such as energy intake and expenditure, macronutrient proportion, food selection patterns, eating habits, and lifestyles play important roles in the formation of body composition and metabolic status [10,11,12,13,14,15,16,17,18]. Effective weight control in adolescents can reduce the risk of developing metabolic diseases commonly observed in adulthood. Recent studies have demonstrated that the composition of macronutrients in the diet and food selection patterns affect body fat and muscle composition and endocrine metabolism in children and adolescents [12,13,14,15,16,17,19]. However, several other studies reported that bodyweight reduction can be accomplished regardless of dietary macronutrient composition, and individualized dietary intervention with decreased total energy intake to achieve optimal compliance may be the key for proper weight reduction [20,21,22,23,24].

Changes in body composition such as body fat mass (BFM), fat-free mass (FFM), and the ratio between these two factors in adolescence by gender, the time period in which secondary sexual characteristics begin to develop, can alter hormonal and metabolic status, and consequently can affect the health status and risk of developing chronic diseases in adulthood [4,5,6]. Modulation of diet composition can affect the secretion of endogenous hormones related to satiety, as well as glucose and lipid metabolisms [14,15,25,26,27]. In addition, it was reported that maternal pre-pregnancy and life course nutrition rather than nutrition during pregnancy itself is more clearly linked to offspring fetal nutrition and growth, as well as bodyweight [28,29]. That is, fetal nutrition can be influenced by the mother’s own early life nutrition, including the accumulated energy balance of the mother in the months and years prior to conception [28,29]. In summary, health and nutritional status during childhood and adolescence, which affect probable body composition and health status in adulthood, are expected to influence the body composition and metabolic status of future offspring.

Therefore, we reviewed articles including observational studies (i.e., cross-sectional, intervention and longitudinal studies) with the key words such as children and adolescent obesity, body composition, macronutrients, eating habits, maternal nutrition, off-spring published in English since 1990 through Pubmed (http://pubmed.ncbi.nlm.nih.gov). We also considered systematic reviews and meta-analysis performed in adults as well as children and adolescents. This narrative review aimed to analyze the role of dietary patterns, particularly of macronutrient intake in growth, body composition and metabolic changes in adolescents, and thereby providing novel insight into the establishment of optimal dietary guidelines for healthy growth with accretion of adequate body composition in the adolescent.

## 2. Trends in Obesity and Related Metabolic Disease in Childhood and Adolescence

Obesity in childhood and adolescence has become a major global public health concern [3,4]. According to the previously mentioned WHO report, one out of ten children and adolescents are obese, and this proportion is rapidly increasing in many parts of the world [1,2,4]. In particular, a higher prevalence of overweight and obesity has been reported among younger adolescents and boys, and the average prevalence of overweight and obesity was 19% across countries and regions included in the WHO survey of health behavior in school-aged children (HBSC), with the highest levels of overweight and obesity generally being in Southern European countries [4]. Based on data from the 2015–2016 National Health and Nutrition Examination Survey (NHANES) in the United States, the prevalence of obesity was 18.5% and affected approximately 13.7 million children and adolescents [30]. In that report, the prevalence of obesity was higher among youth aged 6 to 11 years (18.4%) and adolescents aged 12 to 19 years (20.6%) compared to the prevalence among children aged 2 to 5 years (13.9%) [30]. According to the 2017 report from the Organization for Economic Co-operation and Development (OECD) [31], the average ratio of overweight and obesity in boys was 25.6%, with higher ratios in Chile (46.2%), USA (41.3%), and New Zealand (32.8%). The incidence of childhood and adolescent obesity in the Republic of Korea increased steadily from 11.2% in 2008 to 25% in 2018; furthermore, the rate of increase for severe obesity is much higher (137.5%) than for moderate (30.6%) and mild (57.1%) obesity during the same period [32,33]. The prevalence of severe obesity is also increasing rapidly among Korean young adults aged 20 to 30 years, which may be due to the rapid increase in adolescent obesity during the last 20 years [32]. Moreover, four of every five obese adolescents are at risk for chronic diseases frequently observed in obese adults (i.e., type 2 diabetes, asthma, sleep apnea, and cardiovascular disease), as well as emotional (psychological) and social (school absenteeism and social isolation) problems [4,5,6]. Therefore, optimal strategies to reduce obesity are critical for slowing down and reversing the observed trends of increasing obesity prevalence and its related complications.

## 3. Factors Influencing Adolescent Obesity and Body Composition

Obesity affects adolescents’ quality of life and can result from various factors, either alone or in combination, which include behavioral, emotional, and social factors, as well as genetic background [7,8,9] (Figure 1A,B). In general, obesity occurs due to an imbalance between energy intake and expenditure as a result of unhealthy diet and sedentary lifestyle [34,35,36]. Dietary pattern (i.e., energy intake, macronutrient composition, and food selection), eating habits, and lifestyle (including physical activity level) all play important roles in forming the body composition, including BFM, FFM, the ratio between these two factors, body mass index (BMI), and metabolic status [10,11,12,13,14,15,16,17,18,19,20,21,22,23,24] (Figure 1A). According to data from the WHO HBSC survey, the primary causes of obesity in children and adolescents can be traced to energy-related behaviors (i.e., physical activity, sedentary lifestyle, eating habits, and sleep patterns), which contribute to imbalances between energy intake and energy expenditure [4]. In addition, the report published by the WHO Commission on Ending Childhood Obesity identified obesogenic environments such as lower physical activity, higher sedentary behavior, and intake of energy-dense, nutrient-poor foods as key drivers for the development of obesity [37]. Furthermore, dietary patterns and lifestyle in maternal and infant periods, as well as in childhood and adolescence, may have impacts on health status and obesity-related disease risk throughout the lifetime [28,29] (Figure 1B).

## 4. Association between Dietary Macronutrient Intake Pattern, Obesity, and Related Metabolic Status in Children and Adolescents

Many interventions have been attempted to reduce bodyweight, maintain healthy bodyweight, and improve cardiometabolic risk factors by controlling dietary factors; for example, by reducing total energy intake, changing dietary macronutrient composition, and increasing energy expenditure [10,11,12,13,14,15,16,17,18,19,20,21,22,23,24]. Here, we summarize the effects of dietary pattern on body composition and metabolic status among children and adolescents, with a focus on dietary macronutrient modulation (DMM).

### 4.1. Definition of Low- or High-Macronutrient Dietary Compositions

Different studies apply a wide variety of definitions for low and high dietary composition of macronutrients. For example, macronutrient recommendations from 5 global institutions and governments (USA, Europe, Norway, Australia, New Zealand) have defined a balanced diet as one in which the proportions of total energy intake derived from carbohydrates, fat, and protein are between 45% and 65%, 25% and 35%, and 10% and 20%, respectively [39,40,41,42,43]. This means that a carbohydrate intake of over 65% of the total energy intake indicates high carbohydrate consumption, while a fat intake lower than 25% of the total energy intake indicates low fat consumption in the diet. Furthermore, the Akins diet, one of the most famous low-carbohydrate and high-fat diets, has an extremely restricted carbohydrate intake (5–20% of the total energy intake), and does not limit fat intake regardless of fat type (i.e., saturated or unsaturated) [44,45]. Another famous diet, named the zone diet, defines macronutrient dietary composition differently, promoting low carbohydrate (40%), high protein (30%), and high fat (25–35%) intake; this diet is also concerned with the quality of food ingredients consumed [46,47]. On the other hand, the Current Dietary Reference Intake for Korea (KDRI) defined a balanced diet as one composed of between 55% and 65% carbohydrates, 15% and 30% fats, and 7% and 20% protein; this definition differs from the Western definition [48]. The outcomes of studies can, therefore, be influenced by the definition used in the study. Therefore, this should be taken into consideration when drawing conclusions or comparing different studies.

### 4.2. Beneficial Effect of Dietary Macronutrient Modulation (DMM) on Body Composition and Metabolic Outcomes

Recently, the dietary macronutrient proportion in the diet and food selection patterns of children and adolescents were reported to affect body composition and metabolic status [10,12,13,14,15,16,17,19]. A short-term (12-week) randomized clinical trial for weight loss performed in obese adolescents showed that both a low-fat diet (55% carbohydrate; 20% protein; 25% fat) and modified-carbohydrate diet (35% carbohydrate; 30% protein; 35% fat) significantly improved body composition indicators, such as bodyweight, BMI, waist circumference, and body fat percentage, as assessed by bioelectrical impedance analysis and compared with a wait list control diet [17]. However, the improved body composition levels were not significantly different between the low-fat diet group and modified-carbohydrate diet group. In addition, the significant metabolic improvements were differently observed in the low-fat diet group (insulin resistance, total and LDL-cholesterol, and C-reactive protein) and in the modified-carbohydrate diet group (adiponectin and interleukin-6) [17]. On the other hand, a systemic review of 13 childhood studies compared low-carbohydrate, ad libitum diets with low-fat, energy-restricted diets and reported that the former were more effective for both weight loss and lowering of blood lipid levels at 6 months [10]. A meta-analysis of studies conducted among youth aged 6 to 18 years old also indicated that a low-carbohydrate diet showed a greater reduction in BMI immediately after dietary intervention than did a low-fat diet, even though cardiometabolic benefits from both diets were inconsistent [12].

Many studies have reported the effectiveness of low-carbohydrate and increased protein diets on weight reduction, weight maintenance, and improved cardiometabolic risk in obese adults [11,49,50,51,52,53,54]. Dietary protein is often considered as an important nutrient for diet-based weight management because it modulates neuro-endocrine signals related to satiety, thereby having a higher satiating effect than dietary carbohydrate and fat [6,15,55,56]. Moreover, a high-protein diet showed greater attenuation of a rebound of the relative change in ghrelin over time compared with a high-carbohydrate diet. This was observed in younger children [25] as well as in adults [38,54], although no effects were observed in another study [26]. In fact, sufficient protein intake and physical activity are important for forming body composition, particularly FFM [57]. Protein intake was significantly associated with BMI and FFM in young adults aged in their 20′s [57], increased FFM in children and adolescents [58], and the maintenance of FFM in elderly people [59]. According to Rayman et al. [60,61], high-protein diets exceeding 25% of total energy intake or as protein intake ranging from 0.8 g/kg bodyweight (recommend dietary allowance, RDA) to 1.2–1.6 g/kg (up to double RDA values) in adults improve body composition by preserving lean mass and reducing BFM, especially when combined with training. In addition, Helms et al. suggested that protein intake in the range of 2.3–3.1 g/kg FFM was appropriate for lean, resistance-trained athletes in hypocaloric conditions in adults [62].

### 4.3. Non-Effectiveness of DMM on Body Composition and Metabolic Outcomes

As mentioned above, many studies have reported a beneficial effect of DMM on body composition and metabolic improvement. However, other evidence also suggests that improved weight status and body composition in overweight or obese children and adolescents can be achieved by reducing total energy in the diet and not by modulating macronutrient composition [20,21,22,23,24]. According to reports by Figueroa-Colon et al. [19] and Demol et al. [21], hypocaloric diets were effective in the treatment of obese adolescents and children, but the weight loss effect was not associated with macronutrient composition (low-carbohydrate–high-fat or high-carbohydrate–low-fat diets). In addition, several studies comparing the weight reduction effect of protein-increased diets in obese children and adolescents found the effect to be not significantly different from that of standard protein diet, regardless of the intervention periods (from 1 month to 2 years) [22,23,24,63]. These studies also showed no significant differences in the improvement of metabolic parameters such as blood pressure, lipid profiles, glucose concentrations, and insulin sensitivity [22,23,24,39]. According to a report by Sunehag et al. [14], obese adolescents had increased insulin secretory demands regardless of diet, and failure to increase insulin sensitivity while receiving a high-carbohydrate diet required a further increase in insulin secretion, which may lead to earlier β-cell failure. In addition, a high-fat (low-carbohydrate) diet could potentially increase the risk of fatty liver in obese adolescents [14]. These findings may indicate that neither the fat nor the carbohydrate composition in the diet have any effect on the improvement of insulin resistance in otherwise healthy adolescents who are obese [14]. Further investigations are needed to elucidate the macronutrient effect on weight loss and metabolic improvement in obese adolescents.

Table 1, presented on the next page, summarizes the descriptive information for the selected studies on the macronutrient effects on body composition and metabolic improvement in obese children and adolescents in terms of beneficial effects and non-effectiveness. More details are explained in Section 4.2 and Section 4.3.

## 5. Additional Factors Determining the Effectiveness of DMM on Body Adiposity and Metabolic Improvement

The effects of DMM-based interventions (i.e., a low-carbohydrate, increased-protein diet) on weight reduction and metabolic improvement in obese children and adolescents were inconsistent in the analyzed studies [10,12,13,14,15,16,17,18,19,20,21,22,23,24,61]. However, based on previous study results [19,21,22,64,65,66], we speculated that if the effects of several elements are taken into consideration, modulating macronutrient intake may improve body composition and specific cardiometabolic risk factors in obese children and adolescents.

### 5.1. Study Period

As with the study definition, the study period (short-term or long-term) should also be considered. For example, the beneficial effects of a low-carbohydrate diet compared with a standard low-fat diet on weight reduction and metabolic improvement in children and adolescents have usually been observed in the short term (10–12 weeks) and not in the long term (1 years or more) [19,21,22]. According to a report by Naude et al. [64], weight loss trials in overweight and obese adults showed weight reduction in the short term, irrespective of whether the diet is low in carbohydrate or balanced, and showed little or no difference in weight loss and changes in cardiovascular risk factors up to two years of follow-up. In fact, the study periods for weight reduction conducted in obese children and adolescents are typically shorter than those in obese adults. Therefore, to elucidate the weight loss effects of various dietary interventions in obese children and adolescents, long-term studies are needed. It should be noted, however, that body growth rate and hormonal changes should also be carefully considered, because secondary sexual characteristics begin to develop during this period. Furthermore, policies with a long-range view are needed in order to implement the optimal changes in the typical diets of children and adolescents to improve their health outcomes in real life.

### 5.2. Typical Eating Habits

Previous studies emphasized the importance of eating habits, such as regularity of breakfast intake and meal time, and the frequency of overeating and eating out in terms of healthy weight maintenance and improvement of metabolic abnormality [65,66]. For example, a randomized crossover trial with two 14-day isoenergetic diet periods separated by a 14-day habitual diet washout period compared the effects of regular (6 meals/day) and irregular (3–9 meals/day) meal patterns on the thermic effect of food (TEF), carbohydrate metabolism, subjective appetite, and gut hormones in healthy women [65]. The study findings showed that a regular meal pattern presented a greater TEF, greater insulin sensitivity, lower glucose responses, and beneficial subjective appetite changes, which may favor weight management and metabolic health. According to the cohort analysis of the National Survey for Health and Development, subjects with a more irregular intake of energy, especially at breakfast and lunch, appeared to have an increased cardiometabolic risk 10 and 17 years later [66].

Taken together, the findings of the studies analyzed in this narrative review suggest that DMM is an important part of therapeutic lifestyle intervention for weight control and healthy metabolic status. However, the effectiveness of DMM may not always be observed during the intervention period due to various influencing or confounding factors in the real world. Therefore, there is a need for persistence in maintaining the lifestyle modifications, including meal regularity and physical activity, over a long period, in order to accomplish the goal of achieving healthy body composition and metabolic status.

## 6. Effects of Adolescent Macronutrient Intake on Body Composition and Metabolic Status in Adulthood and Consequent Effects on the Health of the Next Generation

Previous studies suggested that maternal pre-pregnancy and life course nutrition rather than nutrition during pregnancy itself is more clearly linked to the offspring’s fetal nutrition and growth and bodyweight [28,29]; that is, fetal nutrition can be influenced by the mother’s own early life nutrition, including the accumulated energy balance of the mother in the months and years prior to conception [28,29]. Furthermore, the mother’s own fetal life determines the germinal epithelium, where mature endometrial glands form, implying that the qualities of the endometrium may in part reflect the gestational nutritional conditions of the grandmother [29,67]. Of course, maternal macronutrient intake may influence the development and metabolic status of the offspring [55,56,68]. The Danish Fetal Origins Cohort study demonstrated that higher maternal dietary protein intake at the expense of carbohydrates during the second trimester of pregnancy was associated with a modest increase in offspring blood pressure in young adulthood [55]. On the other hand, maternal protein restriction in animal models throughout pregnancy and lactation, as well as during lactation only, was associated with placental dysfunction, offspring growth retardation, a generalized reduction in organ growth (with brain- and heart-sparing effects), and improved insulin sensitivity in the offspring [56,69,70,71,72,73], however protein restriction during the pregnancy only and not during lactation led to insulin resistance in the male offspring [73]. In addition, a lower protein-to-carbohydrate ratio (high-carbohydrate diet) in the maternal diet during pregnancy is associated with higher systolic blood pressure in childhood, which continues up to 4 years of age [74]. Consumption of diets including high saturated fats or refined sugars during pregnancy or lactation promote the pathogenesis of obesity, metabolic syndrome, and cardiovascular disorders in the offspring [75,76]. According to the Healthy Start study, maternal BMI and overnutrition due to a high-fat diet (30% of total energy intake from fat, with a high proportion of saturated fat) can induce obesity, insulin resistance, and impaired glucose utilization in the mothers [77,78]. These unfavorable maternal metabolic conditions may increase fetal growth and birth weight and expose offspring to the risk of obesity and type 2 diabetes in adulthood [77,78]. As briefly mentioned above, changes in body composition (i.e., the proportion and distribution of body FFM, FM, as well as BMI) in adolescence can alter the hormonal and metabolic status, consequently affecting adulthood health status and the risk of developing chronic diseases in adulthood. Based on the previous studies, we speculated that health and nutritional status during childhood and adolescence, which affect the probable body composition and health status in adulthood, are expected to influence the body composition and metabolic status of future offspring, i.e., the next generation (Figure 1B). In addition, changes in macronutrients ratios, especially in pregnancy, children, and adolescents, must be carried out very carefully and with contrasted scientific evidence, since this population is much more sensitive to changes of macronutrient proportions. Further studies are needed to clarify the relationship between adolescent health status and the health status of the next generation.

## 7. Summary

This study reviewed the effects of DMM on body composition and metabolic changes in adolescents and the consecutive influence on health status in adulthood, as well as on probable body composition and metabolic status in the next generation. Based on the findings of previous and recent reports, we suggest that DMM, including sufficient protein intake with slightly reduced carbohydrate intake, may be the most suitable way to improve body composition and metabolic status in obese adolescents, who should still grow up normally with the development of secondary sexual characteristics. However, this narrative review has some limitation. This is a narrative review that describes and appraises published articles so as to speculate on the new types of interventions and meta-analysis available, but may include potential selection and evaluation biases. This is in contrast to systemic reviews and meta-analyses, whose queries, criteria, and article selection processes are well-defined. In addition, this narrative review focused on the major types of macronutrients. The types of macronutrient subgroups (i.e., simple sugar, saturated fats) need to be considered in further studies. Despite these limitations, our review may provide novel insights into optimal dietary guidelines for healthy growth with accretion of adequate body composition during adolescence. It also provides guidance for effective strategies for prevention and management of the risk of obesity-related metabolic disease in adulthood and potential impacts on the health of the next generation.

## Figures and Tables

**Figure 1 nutrients-12-03722-f001:**
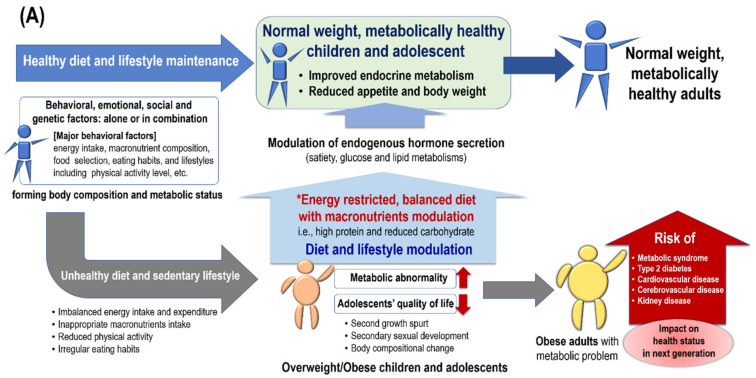
Effect of an energy-restricted, balanced diet with macronutrient modulation on body composition and metabolic status in children and adolescents, as well as throughout the lifetime. (**A**) Effect of dietary macronutrient modulation on children and adolescent body composition and metabolic status. *Energy restricted, balanced diets are effective in the treatment of obese adolescents and children, but the modulation effects of macronutrient composition on weight loss were inconsistent among the studies (**B**) Obesity-related disease risk throughout the lifetime [10,12,13,14,15,16,17,18,19,20,21,22,23,24,38].

**Table 1 nutrients-12-03722-t001:** Descriptive information for the selected studies on the macronutrient effects on body composition and metabolic improvement in obese children and adolescents.

Ref	Sources	Region	Subjects Characteristics	Study Design	Dietary Macronutrient Information	Main Outcome
[14]	J Clin Endocrinol Metab; 90: 4496–4502, 2005	Italy	Healthy obesity (boys *n* = 6; girls *n* = 7, 14.7 ± 0.3 y, BMI: 34 ± 1 kg/m^2^, body fat: 42 ± 1%); normal weight adolescents in previous studies	Randomized crossover design (7-d)	Isocaloric/isonitrogenous diet; High-CHO: 60% CHO, 25% FAT; low-CHO: 30% CHO, 55% FAT	Increased total cholesterol, β-OH butyrate, and gluconeogenesis in the low-CHO group
[15]	Nutrients; 11: 340, 2019	Australia	Obesity (*n* = 8, 16.1 ± 0.4 y, BMI ≥ 95th percentile); healthy weight (*n* = 12, 16.0 ± 0.6 y, BMI: 5–85th percentile)	Double-blind crossover design	High-CHO: 79% CHO, 5% PRO, 16% FAT. High-PRO: 55% PRO, 30% CHO, 15% FAT	Higher ghrelin reduction in high-PRO group than high-CHO group; no differences in postprandial GLP-1 and PYY between the groups; higher postprandial glucose and insulin in high-CHO group than high-PRO group, particularly among obese subjects than normal weight subjects
[16]	Eur J Nutr; 58: 2327–2333, 2019	Australia	Obesity (*n* = 13, 15.4 ± 0.5 y, BMI ≥ 95th percentile); healthy weight (*n* = 13, 16.0 ± 0.6 y, BMI: 5–85th percentile)	Double-blind crossover design	High-CHO: 79% CHO, 5% PRO, 16% FAT. High-PRO: 55% PRO, 30% CHO, 15% FAT	Higher meal-induced thermogenesis in high-PRO group than high-CHO group; higher hunger in high-CHO group than high-PRO group
[17]	PLoS ONE; 11: e0151787, 2016	Australia	Hospitalized overweight /obesity (10–17 y, BMI ≥ 90th percentile): SMC (*n* = 37, 13.2 y), SLF (*n* = 36, 13.2 y), control (*n* = 14, 13.6 y)	Randomized, clinical trial (12-wk)	SMC: 35% CHO, 30% PRO, 35% FAT; SLF: 55% CHO, 20% PRO, 25% FAT	BMI z-score improvement in both SMC and, SLF groups; improvement of insulin resistance, total and LDL-cholesterol, and C-reactive protein in SLF group; improvement of adiponectin and interleukin-6 in SMC group
[19]	Am J Dis Child; 147: 160–166, 1993	USA	Obesity (*n* = 19, 7.5–16.9 y)	Intervention (14.5-m)	PSMFD: 2520 to 3360 J; HBD: 3360 to 4200 J	Significant weight reduction in both diet groups after 6 months; more weight reduction and fat-free mass maintenance in PSMFD group at 10-wk and 6-m
[21]	Pediatr Obes; 12; 179–194, 2017	Israel	Health obesity (boys *n* = 21, girls *n* = 34); 12–18 y (14.4 ± 1.7 y); BMI ≥ 90th percentile	Randomized clinical trial (12-wk) and follow-up (9-m)	LCLF (*n* = 18): 60g CHO (up to 20%), 30% FAT, 50% PRO; LCHF (*n* = 17): 60 g CHO (up to 20%), 20% FAT, 60% PRO; HCLF (*n* = 20): 50–60% CHO, 30% FAT, 20% PRO	No differences in the changes of BMI, BMI percentile, body fat %, and metabolic parameters among the all groups; reduction of insulin and insulin resistance in LCLF and LCHF groups
[22]	Obesity; 17;1808–1810, 2009	USA	Healthy obesity (boys *n* = 34, girls *n* = 61, 9–18 y)	Randomized trials (31-d)	SD (*n* = 49): 50–55% CHO, 15% PRO, 30–35% FAT; High-PRO (*n* = 46): 40–45% CHO, 25% PRO, 30–35% FAT	Reduction of BMI (−2 ± 0.25 kg/m^2^) and other anthropometric parameters except body fat and systolic blood pressure in both groups; no differences of height or appetite between the two groups; increased desire for appetite in both groups
[23]	Am J Clin Nutr; 97; 276–285, 2013	Spain	Healthy obesity (boys *n* = 58, girls *n* = 55, 7–15 y), BMI ≥ 90th percentile	Randomized clinical trial (12-wk) and follow-up (2-y)	LGD (*n* = 57): 45–50% CHO-LG), 20–25% PRO, 30–35% FAT; LFD (*n* = 56): 55–60% CHO-no LGI, 15–20% PRO, 25–30% FAT	Significant reduction of BMI z-score in both groups; no significant differences in the changes of weight loss, insulin resistance, and metabolic syndrome risk factors between LGD and LFD groups
[24]	J Clin Endocrinol Metab; 98; 2116–2125	Australia	Overweight/obesity with prediabetes or insulin resistance (*n* = 111, boys 45, girls 66, 10–17 y)	Randomized clinical trials (12-m) and follow-up (24-m) *	HCLF (*n* = 55): 55–60% CHO-MGL, 30% FAT (≤10% saturated fat), 15% PRO; MCIP (*n* = 56): 40–45% CHO-MGL, 30% FAT(≤10% saturated fat), 25–30% PRO	Increased insulin sensitivity and reduced insulin to glucose in MCIP after 6-m; significant reductions of fasting insulin, SBP z-score, and DBP z-score in HCLF and MCIP groups, but not different between the two groups
[63]	Intern J Obes; 28; 514–519, 2004	France	Healthy obesity (boys *n* = 32, girls *n* = 89, 11–16 y, BMI ≥ 97 percentile)	Intervention (9-m) and follow-up (2-y)	Low-PRO (*n* = 53), 54% CHO, 15% PRO; High-PRO (*n* = 46), 50% CHO, 19% PRO with energy by breakfast 20%, lunch 31%, afternoon snack 16%, and dinner 33%	Reduction of BMI and BMI z-score in Low-PRO and High-PRO groups after the intervention, but yoyo effect observed after 2-y follow-up; no significant differences in the changes of weight loss (BMI z-score) between the two groups

CHO: carbohydrate; d: day, FAT: fat; HBD: hypocaloric balanced diet; HCLF: high-carbohydrate and low-fat diet; LCLF: low-carbohydrate and low-fat diet; LFD: low-fat diet MGL: moderate glycemic load; LGD: low glycemic diet; LGI: low glycemic index; m: month; MCIP: moderate-carbohydrate, increased protein diet %; SD: standard diet; SLF: structured low-fat diet; SMC: structured modified-carbohydrate diet; PRO: protein; PSMFD: protein-sparing modified fast diet; wk: week; y: year; * Metformin consumed during the intervention (start with 500 mg/day and end with 1000 mg/day).

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
