# Peer review of "Impacts of Dietary Macronutrient Pattern on Adolescent Body Composition and Metabolic Risk: Current and Future Health Status—A Narrative Review"

_nutrients, 2020, doi:10.3390/nu12123722_

Round 1

Reviewer 1 Report

The authors present a review regarding the effects of dietary macronutrient patterns on body composition and metabolic changes in adolescents and the consecutive influence on health status in their adulthood, and probable body composition and metabolic status in the next generation. However, it is not sufficiently clear whether modifying the proportions of dietary macronutrients actually has a positive effect to the improve of body composition and metabolic status in obese adolescents (in the long term).

Major issues

  • In a first instance (section 4.2), several positive results on the improvement of body composition and metabolic state are shown. On the contrary, in the following section (4.3) non-effectiveness of DMM on body composition and metabolic outcomes are reported. However, from this information, it seems that no clear conclusion can be reached. In turn, you conclude in lines (190-193): “These findings may indicate that neither the fat nor the carbohydrate composition in the diet have any effect on the improvement of insulin resistance in otherwise healthy adolescents who are obese. Further investigations are needed to elucidate the macronutrient effect on weight loss and metabolic improvement in obese adolescents”.
  • The quality and interpretation of the manuscript would improve considerably with the addition of tables or figures where the effect of different macronutrients on body composition and metabolic status can be clearly shown.
  • It should be noted that the child and adolescent population is much more sensitive to changes of macronutrient proportions. Therefore, at this stage of life, changes in the macronutrient ratios should be made under a contrasted scientific evidence. At the same time, it is necessary to emphasize the importance of evaluating the long-term benefit. It should also be taken into account how this change could be implemented in reality on the typical diet of adolescents.

Minor issues

  • Although the effect exerted by the modulation of macronutrients has been taken into account, the type of subgroups of these has not been considered: i.e. simple sugars, saturated fats, etc. This could also have a great impact on the metabolic state and body composition.
  • Line 149: the low-da diet group??
  • Line 150: cholesterols??
  • Line 393, 395, etc.: double numbering
  • Line 425 and following: double numbering

Author Response

Answers for Editor and Reviews’ comments

Manuscript ID: nutrients-987590.R1

Title: Impact of dietary macronutrients pattern on adolescent body composition and metabolic risk: current and future health status

Nutrients

We sincerely appreciate the time spent in reviewing this manuscript and your advice to improve it.

Please, see below our answers to your queries and comments. We also marked the corrected and revised parts of the text with red highlight. We hope that you find them satisfactory.

Reviewer 1’s comments: The authors present a review regarding the effects of dietary macronutrient patterns on body composition and metabolic changes in adolescents and the consecutive influence on health status in their adulthood, and probable body composition and metabolic status in the next generation. However, it is not sufficiently clear whether modifying the proportions of dietary macronutrients actually has a positive effect to the improve of body composition and metabolic status in obese adolescents (in the long term).

Major issues

In a first instance (section 4.2), several positive results on the improvement of body composition and metabolic state are shown. On the contrary, in the following section (4.3) non-effectiveness of DMM on body composition and metabolic outcomes are reported. However, from this information, it seems that no clear conclusion can be reached. In turn, you conclude in lines (190-193): “These findings may indicate that neither the fat nor the carbohydrate composition in the diet have any effect on the improvement of insulin resistance in otherwise healthy adolescents who are obese. Further investigations are needed to elucidate the macronutrient effect on weight loss and metabolic improvement in obese adolescents”. The quality and interpretation of the manuscript would improve considerably with the addition of tables or figures where the effect of different macronutrients on body composition and metabolic status can be clearly shown.

Answer) The authors sincerely appreciate the reviewer’s advice for improving the manuscript. In accordance with your advice, we added Table 1 on the descriptive presentations of selected major studies for the macronutrient effect on weight loss and metabolic improvement in obese adolescents to improve the interpretation of our manuscript. (Table 1. Descriptive presentation of the selected studies for the macronutrient effect on weight loss and metabolic improvement in obese adolescents.)

It should be noted that the child and adolescent population is much more sensitive to changes of macronutrient proportions. Therefore, at this stage of life, changes in the macronutrient ratios should be made under a contrasted scientific evidence. At the same time, it is necessary to emphasize the importance of evaluating the long-term benefit. It should also be taken into account how this change could be implemented in reality on the typical diet of adolescents.

Answer) Thank you again for your comment. As you pointed out, changes of macronutrient proportions in childhood and adolescence where body growth and secondary sexual characteristics begin to develop should be carefully made based on many of contrasted scientific evidences including the evaluation of the long-term benefit of changes in macronutrient proportions at this stage of life. Furthermore, the policies must take a long-range view to implement the optimal changes in typical diet of adolescents for their health outcome in real life.

Therefore, we added Table 1 summarizing the effectiveness or non-effectiveness of dietary macronutrient modification (DMM) on body composition and metabolic outcomes during childhood and adolescence from the selected papers. We also mentioned the necessity of evaluating long-term benefit of macronutrient proportions in childhood and adolescence mentioned the in section ‘5. 1. Study period’, and the policies with a long range view to implement the optimal changes in typical diet of adolescents for their health outcome in real life.

Minor issues

Although the effect exerted by the modulation of macronutrients has been taken into account, the type of subgroups of these has not been considered: i.e. simple sugars, saturated fats, etc. This could also have a great impact on the metabolic state and body composition.

Answer) In this narrative review, we focused on the major type of macronutrients. As you commented, type of macronutrient subgroups (i.e. simple sugar, saturated fats) needs to be considered. We mentioned in the discussion part for further study.

Line 149: the low-da diet group?? Line 150: cholesterols??

Answer) The authors are sincerely sorry for making the reviewer confused. We corrected the typos errors.

the low-da à the low-fat, cholesterols à cholesterol

Line 393, 395, etc.: double numbering, Line 425 and following: double numbering

Answer) We deleted double numbering and revise it.

Reviewer 2 Report

Thank you for the opportunity to review this well-written and timely paper. I have minor comments to improve the paper and clarity of conceptual model.

-Double check references, it seems some are double numbers

-The figure seems to be condensed and are not clear. This may be a formatting issue or pixel problem.

Abstract

-“ It may provide novel insights”- suggest change to “These findings provide insight” 

Part 1

-Line 58/59 “Many studies suggest…” This sentence seems out of place. Could you frame it in the context of adolescence or adult obesity? So women with obesity may then influence the development of future off spring?

-Rationale for cross-sectional and interventional, why not longitudinal studies? Has this already been reported? do you mean observational studies?

-This review also includes systematic reviews throughout. Was this narrative review supplemented by systematic reviews? Should clarify.

Part 4.2

- “carbohydrate; 30% protein; 35% fat) significantly, but similarly improved body composition such as body weight, BMI, waist circumference, and bodyfat percentage assessed by bioelectrical impedance analysis compared with a wait listed control diet.” Clarify "similarly" in this sentence, was the difference non-significant? If not suggest delete "similarly" or rephrase.

-“the low-da diet group” Clarify low-da.

-Suggest breaking the paragraph in 4.2 when start discussing protein components (around 156/157) as it detracts from the carbohydrate and fat comparisons.

-Provide context for reference 58, 59 and 60. Are these studies in adolescents?

Part 6.

-An additional sentence introduction is needed here as it is not apparently clear why the focus is on maternal macronutrient composition when this review is for adolescent health (who are likely not moms yet).

-Clarify use of Figure 1 here, is this Figure 1b?

Author Response

Answers for Editor and Reviews’ comments

Manuscript ID: nutrients-987590.R1

Title: Impact of dietary macronutrients pattern on adolescent body composition and metabolic risk: current and future health status

Nutrients

We sincerely appreciate the time spent in reviewing this manuscript and your advice to improve it.

Please, see below our answers to your queries and comments. We also marked the corrected and revised parts of the text with red highlight. We hope that you find them satisfactory.

Reviewer 2’s comments: Thank you for the opportunity to review this well-written and timely paper. I have minor comments to improve the paper and clarity of conceptual model.

-Double check references, it seems some are double numbers

Answer) Thank you for your comment for improving our manuscript. We corrected it throughout the manuscript.

-The figure seems to be condensed and are not clear. This may be a formatting issue or pixel problem.

Answer) In accordance with your comment, we revised the figure.

Abstract

-“ It may provide novel insights”- suggest change to “These findings provide insight” 

Answer) We revised it.

Part 1

-Line 58/59 “Many studies suggest…” This sentence seems out of place. Could you frame it in the context of adolescence or adult obesity? So women with obesity may then influence the development of future off spring?

Answer) The authors are sorry for making the reviewer confused with unclear explanation. We intended to mention that health and nutritional status during childhood and adolescence which affect probable body composition and health status in their adulthood is expected to influence body composition and metabolic status of their future off-spring. We revise the sentences to make readers more easily understood below.

“In addition, it was reported that maternal pre-pregnancy and life course nutrition rather than nutrition during pregnancy itself is more clearly linked to the offspring’s fetal nutrition and growth, and body weight [28,29]. That is, fetal nutrition can be influenced by the mother’s own early life nutrition including accumulated energy balance of the mother in the months and years prior to conception [28,29]. In summary, health and nutritional status during childhood and adolescence which affect probable body composition and health status in their adulthood is expected to influence body composition and metabolic status of their future off-spring”

  • Reference 28. Institute of Medicine, Subcommittee on Nutritional Status Weight Gain During Pregnancy, Subcommittee on Dietary Intake and Nutrient Supplements During Pregnancy. Nutrition During Pregnancy: Part I, Weight Gain: Part II, Nutrient Supplements. Washington DC: The National Academies Press, 1990.
  • Reference 29. Gray CA, Bartol FF, Tarleton BJ et al. Developmental biology of uterine glands. Biol Reprod 2001;65:1311–23.

-Rationale for cross-sectional and interventional, why not longitudinal studies? Has this already been reported? do you mean observational studies?

Answer) The authors are sorry for making the reviewer confused with unclear explanation. We revised it more clear below:

“Therefore, we reviewed published articles including observational studies (i.e. cross-sectional, intervention and longitudinal studies), a systemic review and a meta-analysis with the key words such as children and adolescent obesity, body composition, macronutrients, eating habits, maternal nutrition, off-spring to analyze the role of dietary patterns, particularly of macronutrient intake in growth, body composition and metabolic changes in adolescents. This narrative review aimed to provide novel insight into the establishment of optimal dietary guidelines for healthy growth with accretion of adequate body composition in the adolescent.”

-This review also includes systematic reviews throughout. Was this narrative review supplemented by systematic reviews? Should clarify.

Answer) As you pointed out, this is a narrative review which describes and appraises published articles to speculate on new types of interventions and meta-analysis available, but may include potential biases of selection and evaluation, different from a systemic review or a meta-analysis whose query and criteria or the selection of articles from the literature is well defined.

To make it clear, we revised the title of the manuscript to: ‘Impact of dietary macronutrients pattern on adolescent body composition and metabolic risk: current and future health status: narrative review’ Also, we added more explanation on the objective of this narrative review in the introduction, and inserted Table 1 for the summary of the study characteristics and outcomes from main articles cited. In addition, we mentioned the limitation of this narrative review, and future suggestion at the end of the discussion part.

In the introduction:

“Therefore, we reviewed published articles including observational studies (i.e. cross-sectional, intervention and longitudinal studies), a systemic review and a meta-analysis with the key words such as children and adolescent obesity, body composition, macronutrients, eating habits, maternal nutrition, off-spring to analyze the role of dietary patterns, particularly of macronutrient intake in growth, body composition and metabolic changes in adolescents. This narrative review aimed to provide novel insight into the establishment of optimal dietary guidelines for healthy growth with accretion of adequate body composition in the adolescent.”

In the main body:

Table 1 summarized the descriptive information of the selected studies for the macronutrient effect on body composition and metabolic improvement in obese children and adolescents in aspect of beneficial effect and non-effectiveness. More details were explained in the section 4.2 and 4.3

In the discussion:

However, this study has some limitation in this review. This is a narrative review which describes and appraises published articles to speculate on new types of interventions and meta-analysis available, but may include potential biases of selection and evaluation, different from a systemic review or a meta-analysis whose query and criteria or the selection of articles from the literature is well defined. In addition, this narrative review focused on the major type of macronutrients. The type of macronutrient

Part 4.2

- “carbohydrate; 30% protein; 35% fat) significantly, but similarly improved body composition such as body weight, BMI, waist circumference, and body fat percentage assessed by bioelectrical impedance analysis compared with a wait listed control diet.” Clarify "similarly" in this sentence, was the difference non-significant? If not suggest delete "similarly" or rephrase.

Answer) As you pointed out, we deleted ‘similarly’ from the sentence and rephrased the sentence to make the content clear.

-“the low-da diet group” Clarify low-da.

Answer) We corrected the typos errors. (the low-da à the low-fat)

-Suggest breaking the paragraph in 4.2 when start discussing protein components (around 156/157) as it detracts from the carbohydrate and fat comparisons. 

Answer) In accordance with your advice, we separate the paragraph.

-Provide context for reference 58, 59 and 60. Are these studies in adolescents?

Answer) The authors are sorry for making the reviewer confused. As you pointed out, these studies were performed in adults. To make the content clear, we added the words “in adults” in the sentences. 

Part 6.

-An additional sentence introduction is needed here as it is not apparently clear why the focus is on maternal macronutrient composition when this review is for adolescent health (who are likely not moms yet).

Answer) The authors are sorry for making the reviewer confused with unclear explanation. We intended to mention that health and nutritional status during childhood and adolescence which affect probable body composition and health status in their adulthood is expected to influence body composition and metabolic status of their future off-spring. We revise the sentences to make readers more easily understood below:

“Previous studies reported that maternal pre-pregnancy and life course nutrition rather than nutrition during pregnancy itself is more clearly linked to the offspring’s fetal nutrition and growth, and body weight [28,29]. That is, fetal nutrition can be influenced by the mother’s own early life nutrition including accumulated energy balance of the mother in the months and years prior to conception [28,29]. Furthermore, mother’s own fetal life determined germinal epithelium where mature endometrial glands form which imply that the qualities of endometrium may in part reflect the gestational nutritional conditions of grandmother [29,67].”

  • Reference 28. Institute of Medicine, Subcommittee on Nutritional Status Weight Gain During Pregnancy, Subcommittee on Dietary Intake and Nutrient Supplements During Pregnancy. Nutrition During Pregnancy: Part I, Weight Gain: Part II, Nutrient Supplements. Washington DC: The National Academies Press, 1990.
  • Reference 29. Gray CA, Bartol FF, Tarleton BJ et al. Developmental biology of uterine glands. Biol Reprod 2001;65:1311–23.
  • Reference 67. Zaneta M Thayer, Julienne Rutherford, Christopher W Kuzawa. The Maternal Nutritional Buffering Model: an evolutionary framework for pregnancy nutritional intervention. Evol Med Public Health. 2020; 2020(1): 14–27.

-Clarify use of Figure 1 here, is this Figure 1b?

Answer) We indicate Figure 1A and 1B in the section 3, and Figure 1B in the section 6.

“ Obesity affects adolescents’ quality of life and can result from various factors, alone or in combination that include behavioral, emotional and social factors as well as genetic background [7-9] (Figure 1 A and B). In general, ………. and metabolic status [10-24] (Figure 1A).  ………. Furthermore, dietary pattern and lifestyle in maternal and infant period as well as in childhood and adolescence may have impact on health status and obesity related disease risk during lifetime [28, 29] (Figure 1B)”

“ …….. Based on the previous studies, we speculated that the health status in adolescence may have an impact on the probable body composition and metabolic status in the next generation (Figure 1B)…….”

Round 2

Reviewer 1 Report

Thanks for this thorough and detailed revision and reply, which answered and addressed all points of criticism sufficiently.

As a resulting and thus remaining minor point, it may be interesting to mention in the manuscript that changes in macronutrient ratios, especially in pregnancy and children, must be carried out very carefully and under contrasted scientific evidence since this population is much more sensitive to changes of macronutrient proportions.

Author Response

Answers for Editor and Reviews’ comments

Manuscript ID: nutrients-987590.R2

Title: Impact of dietary macronutrients pattern on adolescent body composition and metabolic risk: current and future health status - narrative review

Nutrients

We sincerely appreciate the time spent in reviewing this manuscript and your advice to improve it.

Please, see below our answers to your queries and comments. We also marked the corrected and revised parts of the text with yellow highlight. We hope that you find them satisfactory.

Reviewer 1’s comments

Thanks for this thorough and detailed revision and reply, which answered and addressed all points of criticism sufficiently.

As a resulting and thus remaining minor point, it may be interesting to mention in the manuscript that changes in macronutrient ratios, especially in pregnancy and children, must be carried out very carefully and under contrasted scientific evidence since this population is much more sensitive to changes of macronutrient proportions.

Answer) The authors sincerely appreciate the reviewer’s advice for improving the manuscript. In accordance with your advice, we added it in the section 6.